# Exploring daily time-use patterns: ATUS-X data extractor and online diary visualization tool

Kamila Kolpashnikova[1]*, Sarah Flood[2], Oriel Sullivan[3], Liana Sayer[4], Ekaterina Hertog[1], Muzhi Zhou[1], Man-Yee Kan[1], Jooyeoun Suh[5], Jonathan Gershuny[3]

**1** Department of Sociology, Social Science Division, University of Oxford, Oxford, United Kingdom, **2** IPUMS, University of Minnesota, Minneapolis, Minnesota, United States of America, **3** ESRC Centre for Time Use Research, Social Research Institute, University College London, London, United Kingdom, **4** Maryland Time Use Lab, University of Maryland, College Park, Maryland, United States of America, **5** Department of Economics, American University, Washington, District of Columbia, United States of America

☯ These authors contributed equally to this work.
* kamila.kolpashnikova@sociology.ox.ac.uk

**Data Availability Statement:** Data from IPUMS Time Use are available free of charge to all registered researchers. The IPUMS Time Use system is intended for researchers to be able to

## Abstract

Time-use data can often be perceived as inaccessible by non-specialists due to their unique format. This article introduces the ATUS-X diary visualization tool that aims to address the accessibility issue and expand the user base of time-use data by providing users with opportunity to quickly visualize their own subsamples of the American Time Use Survey Data Extractor (ATUS-X). Complementing the ATUS-X, the online tool provides an easy point-and-click interface, making data exploration readily accessible in a visual form. The tool can benefit a wider academic audience, policy-makers, non-academic researchers, and journalists by removing accessibility barriers to time use diaries.

## Introduction

Time-use data are a powerful source for the analysis of daily lives and human behavior. They record all activities on a 24-hour day in the life of respondents. However, to date, most capabilities of time-use data are underutilized, despite the potential of the data for advancing the understanding of persistent inequalities in time use, including those associated with lower subjective well-being and self-reported health [1–3].

The data enable analyses of daily activities, which can help researchers and policy-makers understand the wide variety of activities that individuals engage in, make comparisons about how people in different population subgroups spend their days, and analyze the rhythms of daily life, including the kinds of activities engaged in at particular times of the day. Because respondents report every activity they engage in on a single day, when the activity happened, how long it lasted, where it occurred, and who else was with them, the possible questions that can be answered with these data extend far beyond a simple accounting of time in major activities. They are used to study numerous issues, classically topics such as gender inequality [4], parental investments in children [5], behavior during economic recessions versus times of

select the years of data they want to analyze, the types of time use they want to analyze, and the demographic characteristics by which they want to conduct their analyses. A key feature of the system is the ability to build custom time use variables that summarize the amount of time each ATUS respondent spends in researcher-specified combinations of activities, locations, time of day, and co-presence of others. A brief tutorial on how to use the system is available online (https://www.youtube.com/watch?v=6nGUBfdhOpo&t=67s). There is no need for researchers to download the multiple original files in which these data are stored and to merge them together. The IPUMS Time Use team has performed the necessary data management steps so that researchers can spend more time analyzing their data and less time performing cumbersome and error-prone data manipulations.

**Funding:** The IPUMS Time Use is supported by the Eunice Kennedy Shriver Institute for Child Health and Human Development (R01HD053654). The online tool creation is supported by the European Union's Horizon 2020 research and innovation programme under the Marie Sklodowska-Curie grant (892101). The work of Oxford authors is supported by the European Union's Horizon 2020 research and innovation programme under the Marie Sklodowska-Curie grant agreement No 892101 (awardee: Kamila Kolpashnikova), the John Fell Fund of the University of Oxford No 7609 (awardee: Kamila Kolpashnikova), and European Research Council Consolidator Grant agreement No 771736 (awardee: Man-Yee Kan).

**Competing interests:** The authors have declared that no competing interests exist.

economic growth [6], when and where people work [7], the use of time outside of paid work [8], and time spent with others [9].

More recently, time use research has broadened into areas of public health (in relation to the determinants of both chronic and infectious disease), extensions of national accounting based on accurate estimations of the extent and value of unpaid work, the environmental footprints associated with different activities [10], and well-being research [11, 12]. Moreover, sequences of cross-sectional nationally representative time-use diary studies permit the study of changes in all these behaviors across time. However, these wonderfully rich time-use diaries from the original American Time Use Survey (ATUS) (https://www.bls.gov/tus/home.htm) are complex to analyze, particularly for those who would like to generate statistics about daily routines and well-being without investing in developing a deep understanding of the data.

The difficulties arise in the quick descriptive analysis of time-use data due to its specific format. Time-use data are underutilized because of the complexity of analyzing person-day event-level data. Most analyses default to statistics on group-level averages of total time in daily activities (e.g., mothers' and fathers' total childcare time), rather than leveraging contextual data on the timings of activities, transitions from one activity into another, how interrupted activities are, or the proportions of daily hours covered by certain activities. Nonspecialists in time diary analyses lack the tools to easily manipulate person-day activity records, limiting full use of the data to specialists in academic settings.

In short, the complexity of detailed time use data limits our broader academic and social understanding of daily patterns. It thus constrains the ways how time diary analyses could inform policy on addressing inequities in daily time use, social interaction, and subjective well-being and health.

To respond to this bottleneck in the potential of the time-use data, we have developed a visualization tool that aims to address the accessibility issue and substantially expand the user bases of time-use data. The ATUS-X Diary Visualizer tool makes the time-use data in its detailed form accessible to a wider academic audience as well as to policy-makers, non-academic researchers, and journalists through rich visualizations, a point-and-click interface, and the availability of statistics at one's fingertips. Moreover, it provides the users with ability to visualize their own subsamples of ATUS-X extracts.

The ATUS-X Diary Visualizer tool (http://atusxvisualizer.com) builds on the work of IPUMS Time Use (https://www.atusdata.org/atus/), a collaboration between IPUMS at the University of Minnesota and the Maryland Population Research Center at the University of Maryland, that harmonizes and simplifies the use of the American Time Use Survey (ATUS). The ATUS-X Diary Visualizer tool allows users to create visualizations of IPUMS Time Use data easily. For illustration purposes, the visualization tool uses the subsample for caregivers identified in the ATUS from 2011 to 2019, downloaded from IPUMS [13].

In the following sections, we first provide a brief overview of IPUMS Time Use, then describe the visualization possibilities for ATUS-X data downloaded from IPUMS Time Use, and then illustrate how the resulting visualizations can be interpreted. Using data on caregivers as an illustration, we compare women and men family caregivers. In the Methods section, we discuss the methodology behind the collection of the ATUS data, conversion into IPUMS Time Use format, and the data transformations behind the visualization tool.

## Methods

### Data description

The American Time Use Survey has been collected annually since 2003 by the U.S. Census Bureau for the Bureau of Labor Statistics. IPUMS Time Use simplifies the use of the American

Time Use Survey (ATUS) data through the IPUMS Time Use data extractor ([www.ipums.org/timeuse](www.ipums.org/timeuse)). The original ATUS is a set of time diaries from a cross-sectional sample of the civilian, non-institutionalized U.S. population, drawn from respondents in the Current Population Survey (CPS), the primary U.S. source of labor force statistics. In the ATUS, individuals aged 15 and older are asked to report all activities they engaged in during the 24-hour period from 4 am on the previous day until 4 am on the reporting day in sequential order. Respondents are only interviewed once, but analyses of the time diary data from all respondents provide a representative picture of Americans' time use. The data collected in the time diaries cover important dimensions of daily life, such as paid work, unpaid domestic work, care activities, leisure, sleep, exercise, travel, and volunteering. The American Time Use Survey (ATUS) data contains 210,586 diaries for the period between 2003 and 2019. Updated information on employment status, work hours, and household composition is collected in the ATUS interview two to five months after the outgoing interview for the CPS. Because the ATUS sample is drawn from the CPS, ATUS data can be linked to rich information collected in the CPS for all household members, including labor force participation, household composition (relationship, age, and gender of all household members), and socioeconomic status. ATUS is collected by the Bureau of Labor Statistics for the U.S. Census. ATUS-X data available through IPMS Time Use are secondary, deidentified data.

In addition to the core time diary survey, questions asked of ATUS respondents have expanded over time to include eldercare and periodically include data from topical modules on eating and health, well-being, and workers' paid and unpaid leaves and job flexibilities fielded to a sub-sample of ATUS respondents. The expansion to include eldercare occurred in 2011 when all ATUS respondents were asked whether in the three to four months prior to the interview they provided adult care to anyone who needed help because of a condition related to aging or an existing condition that worsens with age. Elder caregivers also give information about how long the care has been provided, the frequency of care, and the individuals to whom the respondent provides adult care. The elder care questions provide necessary context about how household composition, sociodemographic characteristics, and time constraints are associated with care provision to household and non-household members.

The Eating and Health Module (EHM), funded by the Economic Research Service of U.S. Department of Agriculture, was fielded by BLS in 2006–08 and 2014–16 alongside the ATUS. The Module contains questions on time eating while doing another activity, fast food and soft drink consumption, grocery shopping and meal preparation practices, participation in food assistance programs, exercise, and self-rated health. With the time diary data collected in ATUS, the EHM provides insight into time constraints and household economic conditions affect eating patterns and health.

Well-Being Modules (WBM), funded by the National Institute on Aging (NIA), were fielded with the ATUS by BLS in 2010, 2012, and 2013. The 2021 Well-Being Module, funded by University of Maryland and University of Minnesota with grants from the Eunice Kennedy Shriver National Institute of Child Health and Human Development and the National Science Foundation, will be fielded with the ATUS in March through December. The Well-Being Modules collect information on health and well-being related to life satisfaction, high blood pressure, feeling well-rested, and an overall assessment of the day as well as detailed information about how Americans were feeling at three points during the day on which data were collected. With assessments of well-being directly tied to specific contexts during the day, these data provide unique and rich information about well-being during specific activities as opposed to more general assessments of time in broad sets of activities.

Leave Modules, funded by the U.S. Women's Bureau, were fielded with the ATUS by BLS in 2011 and 2017–2018. The 2011 Leave Module included questions on ability to adjust work

schedules, access to paid and unpaid leave, and self-rated leave. Questions were redesigned for the 2017–2018 Module to collect data on usual work schedules, schedule flexibility, ability to work from home, as well as access to paid and unpaid leave and reasons for taking leave. Questions in the leave modules offer insight into interconnections of economic activities with other activities (e.g. unpaid work, leisure, and sleep).

IPUMS Time Use simplifies access to the rich, complex data in the ATUS and associated modules by eliminating the need to merge the data with the main ATUS files. IPUMS Time Use harmonizes the data for consistency over time, documents changes across time, and delivers the data and documentation via a single online data dissemination system (www.ipums.org/timeuse). Although the coding of ATUS variables is relatively consistent over time, there have been changes in both the categorization of the over 400 daily activities and ATUS survey questions. IPUMS Time Use assigns consistent variable names and codes to variables that change across time and provides detailed metadata displaying original question wording used in each annual data collection. Variable-specific harmonization documentation describes issues of comparability across years of the ATUS. Key features of the system include the ability for researchers to build custom time use variables that summarize time spent in various activities and with others, select only the years and variables they wish to analyze and receive the data in a ready-to-use format.

In contrast, accessing and analyzing data files downloaded directly from BLS is challenging for novice users of time use data. BLS provides nine different files for each year of ATUS data (e.g. 2003–2019 as of this publication) that have different units of analyses (households, persons, and time diary episodes). Creating analysis files requires researchers to determine if they need person or hierarchical data files (e.g. activities nested within persons), to locate and extract variables of interest from the nine data files, and then merge data from the nine files. This same set of steps must be repeated for each single year of ATUS data. Pre-pooled multi-year files are available from BLS, though they lack all of the detail available in the nine separate files available for each year of data. Whether they use single-year or multi-year files, researchers would then still need to harmonize data over time.

Despite the efforts of IPUMS Time Use to simplify access to ATUS data, the data may still be too complex for some to access. The data visualization tool allows an even broader audience—occasional users of time use data and even individuals who aren't in the research community—to ask and answer questions using the ATUS.

### IPUMS time use data quality checks

The IPUMS Time Use team ingests the ATUS data and creates a harmonized data file using software custom built by IPUMS software developers. During harmonization, no detail is lost, but variables are renamed and recoded for consistency over time. Several data quality checks ensure that errors have not been inadvertently introduced during the harmonization process. We verify that each value in the original data has a corresponding value in the harmonized data and that variable-level distributions in the harmonized data are similar to those in the original data. For example, we confirm that the number of men and women in the data file is the same in the original data from BLS and the harmonized data to ensure that we have not made a mistake in the harmonization process. We perform rigorous checks on the time diary data. We confirm that all time diary records are sequential, that the minutes reported in all-time diaries sum to 1440 minutes per day, and that linking keys required for matching activities to persons are robust. In addition, because the IPUMS system builds variables on the fly for researchers, we perform multiple diagnostic tests to ensure that the system is correctly aggregating time across records in the time diary for a representative set of combinations of primary and secondary activities, locations, co-presence selections, and times of the day.

## Visualization

### Transformations for tempograms' and paths' sequences

The main transformations on the original ATUS-X data are the transformation from the long-format diary format to the wide-format sequences for each diary. Taking the performance optimization into account, the sequences of 1-minute time slots (1440 time slots in each sequence) were reduced to 15-minute sequences (96 time slots in total). These transformations are done at the backend of the online tool, using the PHP programming language. The original diaries in the ATUS-X are recorded till the end time of the last activity, so the original length of diaries often extends beyond 24 hours. We capped the final activity to end at 3:59 am, so all diary sequences are of uniform length. Time of the activity, both for start time and end time, is calculated as a floor function:

$$Activity_t = Activity_{\lfloor Minutes/15 \rfloor} \tag{1}$$

where ($t \in [1,96] \cap Z$). Minutes is the recorded clock time transformed into number of minutes from 4 am of the diary day ($Minutes \in [1,1440] \cap Z$).

Then, the resulting wide-format 15-minute-step sequences were used both for tempogram and path visualizations (see Fig 1). The weights of the observations are adjusted using the original ATUS-X weights adjusted to the sample size of 2003–2019 ATUS. The weights for each selected subsample are normalized ($\hat{w}_j = \frac{w_j}{\sum_{k=1}^{n} w_k}$) for the number of observations in the subsample.

### Calculations for totals

For calculation of totals, we summed the number of sequence steps for each activity category in all selected diaries and divided by the sum of diaries' length. For the selected sample of diaries and each $activity_i$, the totals are calculated using the following formula:

$$\sum_{j=1}^{n} \frac{w_j}{\sum_{k=1}^{n} w_k} \frac{\sum_{t=1}^{96} \left\{ \begin{array}{c} 1, \textit{if } diary_{j,t} = activity_i \\ 0, otherwise \end{array} \right\}}{96} = \sum_{j=1}^{n} \frac{w_j}{\sum_{k=1}^{n} w_k} \left( \#\{t=1,2,\ldots,96 | diary_{j,t} = activity_i\} \right) \tag{2}$$

where $diary_{j,t}$ is the activity that the diary $j$ has at the timestamp $t$; n is the number of diaries, and $w_j$ are the adjusted weights for the $j$-th diary. The total shows the proportion of the total

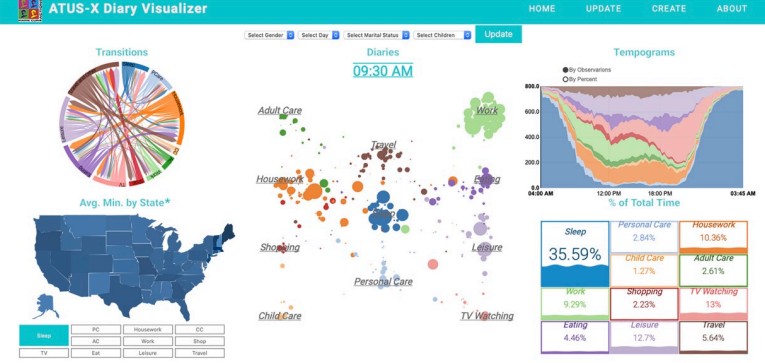

**Fig 1. Overview of the ATUS-X Diary Visualizer online tool.** *The US map was created by tracing a USGS snapshot and is not identical to the original image and is, therefore, for illustrative purposes only. Source: IPUMS ATUS Extractor: Sample of self-identified family caregivers from 2011–2019.

steps in the diaries covered by a certain activity. To display the percentages, the resulting proportions were multiplied by 100.

## Calculation for transitions

For all selected diaries, we counted the total number of activity changes. In other words, we recorded all transitions from $diary_{j,t-1}$ to $diary_{j,t}$ when the activity changed, i.e., $diary_{j,t-1} \neq diary_{j,t}$. For all activity changes, we counted their total number of occurrences and adjusted by the weights of diaries in all steps. The resulting 11 x 11 (rows x columns) matrix represented the weight-adjusted number of transitions from one activity category to another. The categories are: sleep, personal care (PC), housework, childcare (CC), adult care (AC), work, shopping, TV watching, eating, leisure, and travel. Note that we ignored the transitions within the same activity category from t-1 to t, which resulted in the zero diagonal. The weighted transition from activity $u$ into activity $v$ is calculated by the following formula:

$$transition_{u,v} = \sum_{j=1}^{n} \frac{w_j}{\sum_{k=1}^{n} w_k} \left( \#\{t = 2, 3, \ldots, 96 | diary_{j,t-1} = u \text{ and } diary_{j,t} = v\} \right) \quad (3)$$

The resulting proportion shows the weighted proportions of transitions of $activity_{step_{t-1}}$ into $activity_{step_t}$ in the number of all transitions between different activities. To show the percentages, the proportions were multiplied by 100.

## Calculations of state-specific time averages for the US

We denote the total number of diaries from a US state $m$ by $n_m$ and consider them as a single sample. Then, the totals for a given samples are:

$$\sum_{j=1}^{n_m} \frac{w_j}{\sum_{k=1}^{n_m} w_k} \frac{\sum_{t=1}^{96} \left\{ \begin{array}{l} 1, \text{if } diary_{j,t} = activity_i \\ 0, \text{otherwise} \end{array} \right\}}{96} = \sum_{j=1}^{n_m} \frac{w_j}{\sum_{k=1}^{n_m} w_k} \left( \#\{t = 1, 2, \ldots, 96 | diary_{j,t} = activity_i\} \right) (4)$$

This formula represents number of timestamps for a US state $m$ in activity $i$. To get the amount of time spent in activity $i$, we multiply the quantity by the duration of each timestamp, i.e., by 15. We get that the weighted average time spent in activity $i$ is:

$$15 \sum_{j=1}^{n_m} \frac{w_j}{\sum_{k=1}^{n_m} w_k} \left( \#\{t = 1, 2, \ldots, 96 | diary_{j,t} = activity_i\} \right) \quad (5)$$

The resulting average time approximates the average number of minutes spent on the activity based on the observations recorded in the state.

## Code availability

The data visualization tool employs the IPUMS ATUS-X Time Use data based on the samples that users have requested from the IPUMS system. Step-by-step instruction videos are available in this online link (http://atusxvisualizer.com/instructions).

The codes for the online tool can be made available to researchers and developers from a private GitHub repository upon reasonable request, but restrictions apply to the availability of these codes, which were used under different licenses (please check the license details for the JavaScript code snippets in the repository. The PHP code in the back-end controller is the intellectual property of the lead author).

## Results

### Overview of the online tool capabilities

This section describes the *ATUS-X Diary Visualizer* online tool and its main capabilities (http://atusxvisualizer.com). Fig 1 shows an overview snapshot of the tool. Five main visualization plots provide descriptive analyses of ATUS-X data from IPUMS Time Use. By default, the tool displays a sample of self-identified family caregivers from 2011–2019. However, users can upload their own IPUMS ATUS-X extracts using the 'CREATE' button in the navigation bar. We provide more detail in the user guide for the tool (http://atusxvisualizer.com/instructions).

The tool visualizes a random sub-sample of 800 observations in all graphs (for consistency, we used a seeder with the random sample generator, which shows the same 800 observations each time). With the online tool, users can use the dropdown options to choose a subsample of individuals for their analysis. Selections may be based on gender, weekday or weekend diaries, marital status, and parental status. More than 400 detailed activity categories in the ATUS are categorized into eleven main types: sleep, personal care (PC), housework, childcare (CC), adult care (AC), work, shopping, TV watching, eating, leisure, and travel. The full codebook for activity categories is available on the online tool instruction page. The five main graphs in the tool are: 1) *Transitions* plots the number of transition from one activity into another as a percentage of all transitions between activities shown in the highlighted path (disregarding the transitions within the same category of activities); 2) *Diaries (Path Visualization)* –this graph draws the daily paths of the sample sequences, visualizing every 15th minute in the diary starting from 4 am and finishing at 3:45 am; 3) *Tempograms*—these graphs show the number or percent of observations in each activity for each time slot in between 4 am and 3:45 am of the next day; 4) *Average Minutes by State* graph summarizes the average time for the selected activities by each state; and 5) *Percent of Total Time*—this graph summarizes the percent of the total time in all diaries devoted to each activity. In the next sections, we illustrate how the visualizations can facilitate exploratory analysis of time-use data. All graphs plot weighted sequences and totals. Weights are adjusted to the total sample size of the ATUS 2011–2019.

### Transitions

Fig 2 shows the differences between women and men in three activity transitions. Among all transitions for female caregivers, more transitions are from eating to housework (3.19%) than from eating to watching TV (1.61%). The reverse is true for male caregivers—more transitions among men are from eating to watching TV (2.82%) than from eating to housework (2.03%). More men watch TV after eating than do the dishes (or other housework). More women do the dishes (or other housework) after eating than watch TV. These particular transitions are consistent with research showing that women do more housework than men and men enjoy more leisure than women [14, 15].

However, the visualization allows demonstrating how women's household work–and men's leisure time–is paired with other activities in ways that confirm qualitative studies showing women's second shift means their daily time is more fragmented and interrupted by housework and child care, patterns linked with higher stress. Transitions can reveal the common patterns of the activities that happen before or after an activity in focus. For instance, the largest proportions of transitions into eldercare and out of it is taken up by travel. This indicates that many American family caregivers travel to care for the elderly, rather than co-reside with them.

### The use of tempograms

Tempograms in Fig 3 show the daily routines of a sample of female and male family caregivers. It shows that a higher percentage of women spend time on housework during the daytime

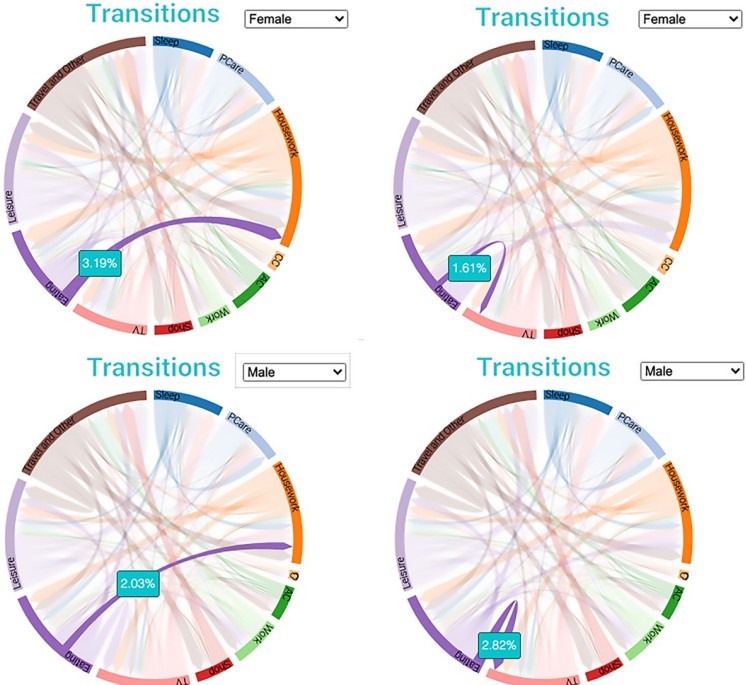

**Fig 2. Diary activity transitions from 1) eating → housework (left) and 2) eating → TV (right) among women (top) and men (bottom).** Source: IPUMS ATUS Extractor: Sample of self-identified family caregivers from 2011–2019.

than men (the area covered by the housework activities is larger on the left-side graph than on the right-side graph). Most housework literature confirms the same for the general population [15–17]. Analogously, a higher percentage of men spend time watching TV in the evenings and during the daytime than women on an average day. This is an interesting observation considering that the previous research suggests that the gender relationship with TV watching is reverse in the general population [18], although supporting results are also present [19].

Additionally, similar to the general population, the tempograms for family caregivers demonstrate that a higher percentage of male caregivers spend time on paid work activities than women caregivers, though the caregivers' sample report less paid work than the general population.

To see single-activity tempograms, the user must click on the activity of interest. For instance, Fig 4 shows the number of women (left) and men (right) performing adult care (top) and childcare (bottom) at every 15-minute interval of the day. The distributions' shapes and

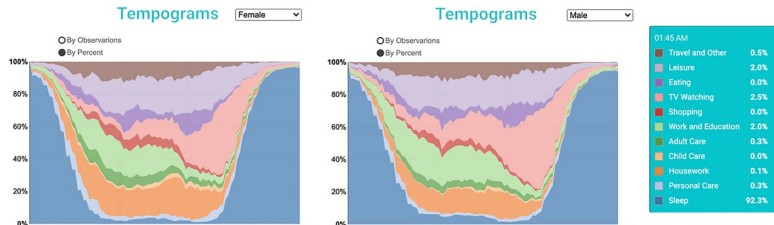

**Fig 3. Tempograms of time use diaries for women (left) and men (right) caregivers.** Source: IPUMS ATUS Extractor: Sample of self-identified family caregivers from 2011–2019.

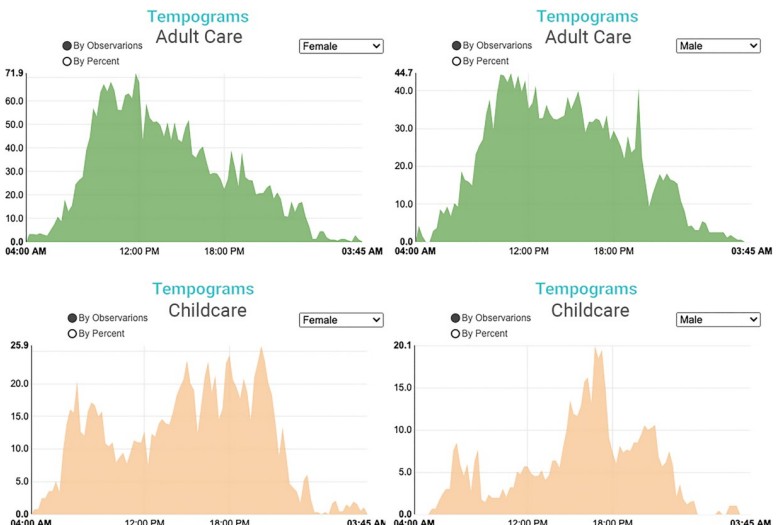

**Fig 4. Adult care time use tempograms for women (top left) and men (top right) and child care for women (bottom left) and men (bottom right).** Source: IPUMS ATUS Extractor: Sample of self-identified family caregivers from 2011–2019.

the total number of observations (out of the initial weighted 800 random observations) are similar among women and men for adult care. In contrast, for childcare, the distributions are denser, and numbers are higher among women than men caregivers. The figure suggests that more women than men are likely to provide childcare, especially during the daytime. The figures also illustrate that adult care is likely to peak in the morning hours, whereas childcare is higher in the evenings. This may be because more support is available for childcare (including nurseries and schools) during the daytime compared to supports for adult care in the US. The lower likelihood of adult care in the evenings suggest that many caregivers might not co-reside with the individuals to whom they provide care. Additionally, among caregivers, more women than men provide adult care at any time during the day, and adult care is more common than childcare [20]. Research shows that women providing adult care are more likely to leave the labor force, which might explain the existing gender differences. The sequencing of care work in this way provides useful knowledge on time incompatibilities between paid work and care work, suggests difficulties in outsourcing care work, as well as the intensity of care work during the day time.

## Activity percent of the total time

Fig 5 describes how total time is distributed separately for all men and women in the sample who provide adult care and who have one or more children under 18 in the household (i.e., sandwich caregivers). Although women spent a little more time on sleep, they spent considerably less time than men watching TV (7.26% vs. 8.11%) and on general leisure activities (9.64% vs. 11.49%). Women also spend less time on paid work (10.47% vs. 16.69%) and more on housework (12.21% vs. 8.63%) and childcare (5.91% vs. 2.8%). The gender differences among sandwich caregivers reflect the gender disparities in the general population, and it is well reported in the literature [21]. Both women and men who are sandwich caregivers spend similar amounts of time on adult care and travel. The observations from the visualization about adult care also reflect the findings of the existing research [22, 23].

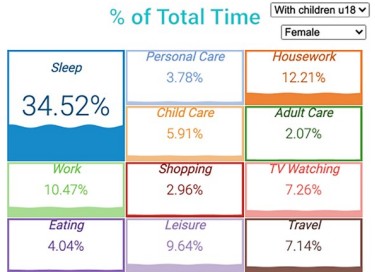

**Fig 5. Percent of total time spent on select activities among women (left) and men (right) sandwich caregivers.**
Source: IPUMS ATUS Extractor: Sample of self-identified family caregivers from 2011–2019.

## Discussion

The development of a quicker way to plot descriptive statistics for time-use data, like the one provided by the ATUS-X Diary Visualizer Tool, can accelerate the initial stages of time-use data analysis for scholars and visualization needs for science communicators and journalists. In such a way, it can potentially facilitate impact assessment for policies affecting time use, notably policies on eldercare, childcare, other domestic work, and employment regulation, especially when such policy has to be developed rapidly.

A prime example would be the recent pandemic, where a populations' sequences of activities (including their locations and co-presence) can be used to throw light on the behavioral determinants of exposure to infection. When the time-use diaries for 2020 are available, their visualization will be made easy by the online tool—the users will be able to upload their subsets of ATUS-X to see if they notice any differences in the different samples, such as contrasting a subsample from 2020 to those of one of the previous years. Time-use data has recently been influential in the area of public health analysis related to chronic health issues [24], so another related potential would lie in the understanding of eating and adult care activities across the day, which are made easy with the tool.

Looking into the future for the tool and its possible uses, a standardized visual look for international comparisons could be of substantial benefit. Most developed countries already collect time-use data. The Multinational Time Use Study (MTUS) is the largest openly accessible nationally representative harmonized time diary survey database, currently with 25 countries represented (https://www.timeuse.org/mtus). Fifteen of these surveys are to date accessible via IPUMS as an MTUS subset (https://www.mtusdata.org/mtus/). Making the ATUS-X extraction and visualization tool compatible with MTUS-X data would allow instant cross-national comparisons and analysis both for exploratory research and policy purposes. As the MTUS-X is episode-based, it will facilitate comparison of sequences of episodes–how time use is distributed over the day–potentially allowing for policymaking to delve into and target specific times of day (e.g., rush hours) and specific activity transitions (for example reflecting transport patterns during rush hours).

Another avenue for future extension of the tool is the use of heriage data, such as AHTUS (https://www.ahtusdata.org/ahtus/) and extending the options to filter data over time. The use of the heritage data and year filters will help historical analysis of the time use data.

## Author Contributions

**Conceptualization:** Kamila Kolpashnikova, Sarah Flood.

**Data curation:** Kamila Kolpashnikova, Sarah Flood, Oriel Sullivan, Liana Sayer.

**Formal analysis:** Kamila Kolpashnikova, Sarah Flood, Oriel Sullivan, Liana Sayer.

**Funding acquisition:** Kamila Kolpashnikova, Sarah Flood, Liana Sayer.

**Investigation:** Sarah Flood.

**Methodology:** Kamila Kolpashnikova, Sarah Flood, Liana Sayer.

**Project administration:** Kamila Kolpashnikova, Sarah Flood.

**Resources:** Kamila Kolpashnikova, Sarah Flood, Oriel Sullivan.

**Software:** Kamila Kolpashnikova, Sarah Flood.

**Supervision:** Liana Sayer, Jonathan Gershuny.

**Validation:** Sarah Flood.

**Visualization:** Kamila Kolpashnikova.

**Writing – original draft:** Kamila Kolpashnikova, Sarah Flood.

**Writing – review & editing:** Kamila Kolpashnikova, Sarah Flood, Oriel Sullivan, Liana Sayer, Ekaterina Hertog, Muzhi Zhou, Man-Yee Kan, Jooyeoun Suh, Jonathan Gershuny.

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
