## [Decision Letter · Decision Letter 0]

20 Mar 2021

PONE-D-21-01724

Exploring daily time-use patterns: ATUS-X data extractor and online diary visualization tool

PLOS ONE

Dear Dr. Kolpashnikova,

Thank you for submitting your manuscript to PLOS ONE. After careful consideration, we feel that it has merit but does not fully meet PLOS ONE’s publication criteria as it currently stands. Therefore, we invite you to submit a revised version of the manuscript that addresses the points raised during the review process.

The ATUS is a great resource, and this manuscript can expand the reach of the data in the research community.  The reviewers have provided detailed feedback on how to increase the clarity of the paper and the reach of the findings. Please address all reviewer comments. There are some interesting suggestions in the last paragraph from Reviewer 1; may be beyond the scope of this paper but important to at least consider and mention as possible directions for future work in this field. 

We look forward to receiving your revised manuscript.

Kind regards,

Solveig A. Cunningham, Ph.D.

Academic Editor

PLOS ONE

Journal Requirements:

4. We note that Figure 1 in your submission contain map images which may be copyrighted. All PLOS content is published under the Creative Commons Attribution License (CC BY 4.0), which means that the manuscript, images, and Supporting Information files will be freely available online, and any third party is permitted to access, download, copy, distribute, and use these materials in any way, even commercially, with proper attribution. For these reasons, we cannot publish previously copyrighted maps or satellite images created using proprietary data, such as Google software (Google Maps, Street View, and Earth). For more information, see our copyright guidelines: http://journals.plos.org/plosone/s/licenses-and-copyright.

4.1.    You may seek permission from the original copyright holder of Figure 1 to publish the content specifically under the CC BY 4.0 license. 

4.2.    If you are unable to obtain permission from the original copyright holder to publish these figures under the CC BY 4.0 license or if the copyright holder’s requirements are incompatible with the CC BY 4.0 license, please either i) remove the figure or ii) supply a replacement figure that complies with the CC BY 4.0 license. Please check copyright information on all replacement figures and update the figure caption with source information. If applicable, please specify in the figure caption text when a figure is similar but not identical to the original image and is therefore for illustrative purposes only.

Reviewers' comments:

Reviewer's Responses to Questions

**Comments to the Author**

1. Is the manuscript technically sound, and do the data support the conclusions?

Reviewer #1: Partly

Reviewer #2: Yes

2. Has the statistical analysis been performed appropriately and rigorously? 

Reviewer #1: N/A

Reviewer #2: Yes

3. Have the authors made all data underlying the findings in their manuscript fully available?

Reviewer #1: Yes

Reviewer #2: Yes

4. Is the manuscript presented in an intelligible fashion and written in standard English?

Reviewer #1: Yes

Reviewer #2: Yes

5. Review Comments to the Author

Reviewer #1: I'm a big fan of the various IPUMS projects, so I enjoyed reading this particular manuscript. I don't think it's ready for publication yet, however. I don't think it suffers from fundamental flaws, but I do think the revisions it needs go beyond minor edits.

I think there should be more of a balance in the manuscript in terms of discussing both the data extraction tool for the ATUS data and the visualization tool. The latter gets more attention, but I think the former is of equal or even greater importance depending on the audience. Researchers who want to do more intensive analyses will find the extraction tool of much greater utility than the visualization tool, as the latter is engaging to use but is limited in terms of what you can do with it (as is often the case with data visualization tools). So, more discussion about the extraction tool is in order, I think.

As part of that discussion, the authors should spend more time discussing the contents of ATUS data in terms of the categories of activities included, the demographic/socio-economic data included, and the content of the topical modules that the manuscript mentions, e.g. on eldercare, health, well-being, etc., so that people have a better sense of how rich the contents of the data are and what opportunities they do (and don't) present. I think that more discussion of the structure of the ATUS data would also be in order in terms of how the data at the BLS are spread across multiple files, as would more discussion of the event-level nature of the data including some visual representation of the diary data. The authors note that the structure of such data is novel to many researchers and thus potentially daunting with which to work. I don't disagree with that, but I also think that the somewhat bare-bones description of the data in the manuscript effectively understates how much more accessible the data extraction tool makes the data. More explanation of how unwieldy the original ATUS data are, and thus how much easier the extraction tool makes it to manage the data, would make that point more clear.

I also have some more specific/particular points. Since the manuscript didn't have page numbers, I'll refer to the line numbers that were included in the manuscript I downloaded.

On lines 81-82, the authors state that "For illustration purposes, the visualization tool uses the subsample for caregivers identified in the ATUS from 2011 to 2019, downloaded from IPUMS." Later, on lines 223-224, they state that users can upload their own extracts for use by the visualization tool; that point should be made earlier, just to make it clear from the beginning that the visualization tool is not confined to the default subset of data.

If I understand correctly, the visualization tool doesn't break results down by time period in terms of year or month or year-month, even though the year and month of the interviews are recorded (see https://www.atusdata.org/atus-action/variables/group/tech_tech). Is my understanding correct? And, if so, what is the basis for the authors' claims on lines 321-324 that the tool will make it possible to visualize the effects of the pandemic on time use, caregiving duties, etc.? I agree that time-use data from 2020 could be useful for studies of, say, how the pandemic did or didn't exacerbate gender inequities in terms of divisions of labor within households. However, I'm not sure what insights can be gathered from a visualization tool that doesn't let you break the data down by dates to see how time allocations shifted during 2020.

Lines 170-172 refer to aggregating diary events data into an 11X11 matrix of categories and transitions between them, but those categories aren't identified until Lines 233-235. The category definitions should be provided earlier, to clarify what sorts of transitions are being visualized.

Lines 237-238 state that "Transitions plots the percent of all transitions from one activity to another shown in the highlighted path." If I understand correctly, the point is that the value for transitions between activity A and activity B is meant to be the number of transitions from A to B as a percentage of all transitions between activities. If so, the phrasing should be tweaked to make that more explicit. As it stands now, the wording is a bit unclear.

Lines 328-333 refer to integrating multinational time-use survey data into the tool, and I agree that such an effort could be very worthwhile. E.G. seeing whether differences in health-care systems and welfare states are associated with different patterns of time use, or different shifts in time use in response to macro-level shocks and crises, could be quite interesting. However, the authors don't mention the heritage time-use data at https://www.ahtusdata.org/ahtus/. Is there any intention of working these data into the visualization tool? Such an integration, in combination with additional features for breaking data down by dates/years, could provide insights with regard to how/whether gender imbalances and household divisions of labor have shifted over the decades as more women participate in the formal labor force. To the extent that there are health implications from certain types of daily activities, as the manuscript suggests, having a tool with longer time ranges of data could likewise be informative with regard to analysis longer-term health trends.

I understand that there could be considerable technical issues with such integration, and I don't expect the authors to resolve them as a condition of publication. :-) But, since they already mention future ideas for the visualization tool, I'd like to see more about extending it to cover the heritage data and provide options to filter/analyze data over time - are such ideas under consideration, would they be feasible, what data-related challenges would they involve, etc.

Reviewer #2: This article introduces the ATUS-X diary visualization tool for time use data. Time use data are certainly underutilized in research, and I suspect one reason why is their complexity. A tool that can help with translation of these data would be very useful. Overall I think there are some improvements that can be made to this manuscript that would help convince the reader that this particular tool will be helpful for visualizing, understanding, and translating time-use data.

Introduction

• Time-use data can be used in many different ways. They may be collapsed into broad categories or used in much finer detail. I think readers who are more recently introduced to the concept could use a bit more clarification on the complexity of time-use data and the various ways in which researchers in many different fields use them. For example, the intro states a few times that these data are ‘powerful’ and ‘wonderfully rich’, but I don’t feel like many examples are given to support those statements. A brief mention of how these data are generally collected may also help readers (on that note- is the visualization tool only useful for ATUS data? Or other time-use data? Only with self-reported diary data, or also accelerometry data, for example).

• It would be helpful to include references for the example studies/study topics of time use (lines 44-47).

Methods

• A reference is needed for the ATUS (BLS website or other).

• As it appears 1 minute epoch lengths are collapsed into 15 minute epoch lengths, it is worth describing the unit of data collection in the ATUS (i.e., do participants report time-use in 1 minute epoch lengths, 10 minute, every second, etc.).

o Related to this, what is the rationale for collapsing time specifically into 15 minute sequences?

Results

• The tempograms can clearly be very useful in a variety of research. I have a bit of trouble seeing the differences in the transitions visualizations and it makes me wonder if there is another way you can describe and display the value of these visualizations. Perhaps this type of figure is just less intuitive and will require a bit more in-text description, or maybe it would be more useful for an analysis with fewer time-use categories? This could be included in the discussion.

• I’m not sure if the figure captions got lost along the way, but if there are none, it seems that adding some in would improve clarity. I also expect the figures will be higher quality in the publication, as they are blurry and not very legible in the PDF.

Discussion

• Overall, I think there needs to be a bit of discussion on what makes the visualizations from this tool better than various tools that already exist. As a person who studies broader categories of time use, I don’t necessarily feel convinced that this tool will allow me to create more intuitive or more translatable figures than those I can create quickly and easily with certain R packages. Why should I make the switch to this tool?

6. PLOS authors have the option to publish the peer review history of their article (what does this mean?). If published, this will include your full peer review and any attached files.

Reviewer #1: No

Reviewer #2: No

---

## [Author Response · Author response to Decision Letter 0]

3 May 2021

RESPONSE MEMO

Reviewer #1: I'm a big fan of the various IPUMS projects, so I enjoyed reading this particular manuscript. I don't think it's ready for publication yet, however. I don't think it suffers from fundamental flaws, but I do think the revisions it needs go beyond minor edits.

COMMENT: I think there should be more of a balance in the manuscript in terms of discussing both the data extraction tool for the ATUS data and the visualization tool. The latter gets more attention, but I think the former is of equal or even greater importance depending on the audience. Researchers who want to do more intensive analyses will find the extraction tool of much greater utility than the visualization tool, as the latter is engaging to use but is limited in terms of what you can do with it (as is often the case with data visualization tools). So, more discussion about the extraction tool is in order, I think.

As part of that discussion, the authors should spend more time discussing the contents of ATUS data in terms of the categories of activities included, the demographic/socio-economic data included, and the content of the topical modules that the manuscript mentions, e.g. on eldercare, health, well-being, etc., so that people have a better sense of how rich the contents of the data are and what opportunities they do (and don't) present. I think that more discussion of the structure of the ATUS data would also be in order in terms of how the data at the BLS are spread across multiple files, as would more discussion of the event-level nature of the data including some visual representation of the diary data. The authors note that the structure of such data is novel to many researchers and thus potentially daunting with which to work. I don't disagree with that, but I also think that the somewhat bare-bones description of the data in the manuscript effectively understates how much more accessible the data extraction tool makes the data. More explanation of how unwieldy the original ATUS data are, and thus how much easier the extraction tool makes it to manage the data, would make that point more clear.

RESPONSE: Following the suggestion, we added an extended discussion of the ATUS extractor together with its methodologies and specific modules as suggested. The ATUS extractor tool is, in fact, a vital part of this project. We added as much information as possible following the suggestions. For example, more explanation on the connections of ATUS with CPS data is added on lines 84-91. The explanation on modules is added on lines 100-128. Specific methodological points are discussed in 129-155.

COMMENT: I also have some more specific/particular points. Since the manuscript didn't have page numbers, I'll refer to the line numbers that were included in the manuscript I downloaded.

On lines 81-82, the authors state that "For illustration purposes, the visualization tool uses the subsample for caregivers identified in the ATUS from 2011 to 2019, downloaded from IPUMS." Later, on lines 223-224, they state that users can upload their own extracts for use by the visualization tool; that point should be made earlier, just to make it clear from the beginning that the visualization tool is not confined to the default subset of data.

RESPONSE: We added an explanation that the visualization tool allows users to visualize their own extract subsamples in the abstract and introduction.

COMMENT: If I understand correctly, the visualization tool doesn't break results down by time period in terms of year or month or year-month, even though the year and month of the interviews are recorded (see https://www.atusdata.org/atus-action/variables/group/tech_tech). Is my understanding correct? And, if so, what is the basis for the authors' claims on lines 321-324 that the tool will make it possible to visualize the effects of the pandemic on time use, caregiving duties, etc.? I agree that time-use data from 2020 could be useful for studies of, say, how the pandemic did or didn't exacerbate gender inequities in terms of divisions of labor within households. However, I'm not sure what insights can be gathered from a visualization tool that doesn't let you break the data down by dates to see how time allocations shifted during 2020.

RESPONSE: At the present iteration of the tool, it is possible by ‘feeding’ two different samples into the tool and analyzing them one by one. However, in the future (provided this project is further funded), the tool’s capabilities as to what it can do will be expanded (for instance, without requiring uploading two different samples). At the moment, the hosting/computational and maintenance work is only supported by the lead author without any specific funding allocated to the development of the project’s part in term of supporting faster computational abilities and data loading speed, and for the development of the online tool by involving more people. So, given the current financial constraints (lead author’s own money), only the current version of the tool can be made available in perpetuity (the lead author will have to continue paying for hosting and domains beyond the current postdoctoral funding). However, this will change over time as soon as the funding is procured. Even without funding, the lead author intends to continue working on financially feasible ways to improve the tool. Among them is to improve responsiveness and speed (probably by migrating to Django and Spark SQL).

COMMENT: Lines 170-172 refer to aggregating diary events data into an 11X11 matrix of categories and transitions between them, but those categories aren't identified until Lines 233-235. The category definitions should be provided earlier, to clarify what sorts of transitions are being visualized.

RESPONSE: We added the category names at the suggested place.

COMMENT: Lines 237-238 state that "Transitions plots the percent of all transitions from one activity to another shown in the highlighted path." If I understand correctly, the point is that the value for transitions between activity A and activity B is meant to be the number of transitions from A to B as a percentage of all transitions between activities. If so, the phrasing should be tweaked to make that more explicit. As it stands now, the wording is a bit unclear.

RESPONSE: We changed the wording as suggested

COMMENT: Lines 328-333 refer to integrating multinational time-use survey data into the tool, and I agree that such an effort could be very worthwhile. E.G. seeing whether differences in health-care systems and welfare states are associated with different patterns of time use, or different shifts in time use in response to macro-level shocks and crises, could be quite interesting. However, the authors don't mention the heritage time-use data at https://www.ahtusdata.org/ahtus/. Is there any intention of working these data into the visualization tool? Such an integration, in combination with additional features for breaking data down by dates/years, could provide insights with regard to how/whether gender imbalances and household divisions of labor have shifted over the decades as more women participate in the formal labor force. To the extent that there are health implications from certain types of daily activities, as the manuscript suggests, having a tool with longer time ranges of data could likewise be informative with regard to analysis longer-term health trends.

I understand that there could be considerable technical issues with such integration, and I don't expect the authors to resolve them as a condition of publication. :-) But, since they already mention future ideas for the visualization tool, I'd like to see more about extending it to cover the heritage data and provide options to filter/analyze data over time - are such ideas under consideration, would they be feasible, what data-related challenges would they involve, etc.

RESPONSE: Thank you very much for pushing us to think more about the even further potentials of the tool. We added the discussion that the heritage data and filters for years could be added to the tool in the future in the discussions part. The implementation of both heritage data and year filters is feasible, and it will be implemented in the future. AHTUS is also harmonized with the MTUS (which allows not only temporal comparisons but also cross-country temporal comparison).

Reviewer #2: 

COMMENT: This article introduces the ATUS-X diary visualization tool for time use data. Time use data are certainly underutilized in research, and I suspect one reason why is their complexity. A tool that can help with translation of these data would be very useful. Overall I think there are some improvements that can be made to this manuscript that would help convince the reader that this particular tool will be helpful for visualizing, understanding, and translating time-use data.

Introduction

• Time-use data can be used in many different ways. They may be collapsed into broad categories or used in much finer detail. I think readers who are more recently introduced to the concept could use a bit more clarification on the complexity of time-use data and the various ways in which researchers in many different fields use them. For example, the intro states a few times that these data are ‘powerful’ and ‘wonderfully rich’, but I don’t feel like many examples are given to support those statements. A brief mention of how these data are generally collected may also help readers (on that note- is the visualization tool only useful for ATUS data? Or other time-use data? Only with self-reported diary data, or also accelerometry data, for example).

RESPONSE: The revised version adds more detail on the ATUS data and the extract in lines 84-155 to explain the time-use data in more detail and what the ATUS contains (e.g., its modules), as well as more detail on its methodological aspects to illustrate both its versatility and richness. At the moment, the visualization tool only works with the ATUS extractor data. The potential to use it on MTUS and heritage data is discussed in the Discussions part. The mention of only two potential applications, of course, does not preclude the potential to be applied to any time use data. However, given the resources that we possess at the moment, we decided to describe only the viable options, which will be implemented even in the case when no additional funding is raised to support the project. Yet, any applications to time use data are possible provided we can secure a big enough team to do so.

COMMENT: • It would be helpful to include references for the example studies/study topics of time use (lines 44-47).

RESPONSE: We added the examples and citations as suggested.

COMMENT: Methods

• A reference is needed for the ATUS (BLS website or other).

RESPONSE: We added citation to BLS on the first mention of ATUS.

COMMENT: • As it appears 1 minute epoch lengths are collapsed into 15 minute epoch lengths, it is worth describing the unit of data collection in the ATUS (i.e., do participants report time-use in 1 minute epoch lengths, 10 minute, every second, etc.).

o Related to this, what is the rationale for collapsing time specifically into 15 minute sequences?

RESPONSE: The decision to reduce information to 15-minutes slots was taken because of the computational ability for the online tool at the moment (for the hosting to continue in perpetuity supported only by the funds of the lead author, it has to be hosted on the servers with lower computational speed). More detailed (such as 1-minute sequence) computations at the (PHP) back end require higher computation power for the hosting servers, which unfortunately we cannot afford at the moment (hosting will have to be paid for by the lead author in perpetuity if no additional funding procured). It is, however, technically possible and will be done when the funding is procured for it in the future. Overall, this is done to increase the speed given the limited power of the server. When the funding allows, the backend will be replaced with Spark SQL / Django configuration with faster server hosting and hopefully with utilizing some acceleration on SQL queries. Basically, many choices in the visualization tool are dictated by the code optimization problem given the limitation of resources. Your suggestion is easily implementable technically and will be implemented when there are enough funds for faster computational/upload speed.

COMMENT: Results

• The tempograms can clearly be very useful in a variety of research. I have a bit of trouble seeing the differences in the transitions visualizations and it makes me wonder if there is another way you can describe and display the value of these visualizations. Perhaps this type of figure is just less intuitive and will require a bit more in-text description, or maybe it would be more useful for an analysis with fewer time-use categories? This could be included in the discussion.

RESPONSE: Transitions show interesting information on the transitions from certain activities and into them, which is underutilized. There are many potentials for the use of transitions data (coupled with event-history methods). For example, researchers could be interested in analyzing how many people actually do something else before work (for example, have breakfast vs people who skip breakfast and head to work right away). This behavior could be analyzed over time. The information would be interesting both from the point of view of public health and for the food industries that aim to provide breakfast meals (e.g., coffee shops). Another example would be among the caregivers sample that is used as the default for the visualization, to see the patterns of activities after work—how frequent are the transitions into eldercare vs transitions into leisure activities etc. 

COMMENT: • I’m not sure if the figure captions got lost along the way, but if there are none, it seems that adding some in would improve clarity. I also expect the figures will be higher quality in the publication, as they are blurry and not very legible in the PDF.

RESPONSE: We updated the figures, added more quality.

COMMENT: Discussion

• Overall, I think there needs to be a bit of discussion on what makes the visualizations from this tool better than various tools that already exist. As a person who studies broader categories of time use, I don’t necessarily feel convinced that this tool will allow me to create more intuitive or more translatable figures than those I can create quickly and easily with certain R packages. Why should I make the switch to this tool?

RESPONSE: The paper stresses that one of the big contributions is broadening accessibility to time use data to people who are unfamiliar with it. Many people who have not used time use data before are likely to benefit from this tool for initial data exploration, even if they can make in the end better visualisations with R. This tool will enable people new to ATUS to identify the most promising avenues for research. In addition, this tool will make time use data easily accessible beyond academia to journalists and policy-makers.

Although we agree that this tool may not be as helpful for someone who is already familiar with time use data and visualization as opposed to a new user, this tool could help researchers who are new to the field to learn more about time use data and data visualization or journalists who could help from easy access to data. We seek to build and support an interdisciplinary and diverse community of researchers using time use survey data in simpler ways.

---

## [Editor Report · Decision Letter 1]

24 May 2021

Exploring daily time-use patterns: ATUS-X data extractor and online diary visualization tool

PONE-D-21-01724R1

Dear Dr. Kolpashnikova,

We’re pleased to inform you that your manuscript has been judged scientifically suitable for publication and will be formally accepted for publication once it meets all outstanding technical requirements.

Kind regards,

Solveig A. Cunningham, Ph.D.

Academic Editor

PLOS ONE

Additional Editor Comments:

Please fix the figures to provide clear titles describing what is being shown, as well as details about the data source and population which the data are describing. The figures are currently not clearly labelled.

---

## [Editor Report · Acceptance letter]

7 Jun 2021

PONE-D-21-01724R1 

Exploring daily time-use patterns: ATUS-X data extractor and online diary visualization tool 

Dear Dr. Kolpashnikova:

I'm pleased to inform you that your manuscript has been deemed suitable for publication in PLOS ONE. Congratulations! Your manuscript is now with our production department. 

Kind regards, 

on behalf of

Dr. Solveig A. Cunningham 

Academic Editor

PLOS ONE